# Three-dimensional structure determination of protein complexes using matrix-landing mass spectrometry

Michael S. Westphall[1], Kenneth W. Lee[1], Austin Z. Salome [2], Jean M. Lodge[1], Timothy Grant [3,4✉] & Joshua J. Coon [1,2,4✉]

Native mass spectrometry (MS) is increasingly used to provide complementary data to electron microscopy (EM) for protein structure characterization. Beyond the ability to provide mass measurements of gas-phase biomolecular ions, MS instruments offer the ability to purify, select, and precisely control the spatial location of these ions. Here we present a modified Orbitrap MS system capable of depositing a native MS ion beam onto EM grids. We further describe the use of a chemical landing matrix that preserves the structural integrity of the deposited particles. With this system we obtain a three-dimensional reconstruction of the 800 kDa protein complex GroEL from gas-phase deposited GroEL ions. These data provide direct evidence that non-covalent protein complexes can indeed retain their condensed-phase structures following ionization and vaporization. Finally, we describe how further developments of this technology could pave the way to an integrated MS-EM technology with promise to provide improved cryo-EM sample preparation over conventional plunge-freezing techniques.

[1] Department of Biomolecular Chemistry, University of Wisconsin-Madison, Madison, WI, United States. [2] Department of Chemistry, University of Wisconsin-Madison, Madison, WI, United States. [3] Department of Biochemistry, University of Wisconsin-Madison, Madison, WI, United States. [4] Morgridge Institute for Research, Madison, WI, United States. ✉email: tim.grant@wisc.edu; coon@wisc.edu

By allowing so-called elephants to fly, electrospray ionization (ESI) coupled with mass spectrometry (MS) has transformed our ability to characterize proteins[1]. Perhaps no area better exemplifies this than native MS, where intact protein complexes are gently ionized, vaporized, and mass analyzed[2]. These mass measurements can provide invaluable information on subunit stoichiometry, connectivity, and even the presence of non-covalently bound ligands.

Whether ionized protein complexes truly retain their structure in the gas phase has been debated for decades[3]. Collisional cross-sections of numerous gas-phase protein complexes have been experimentally determined by ion mobility MS[4–6]. In general, these collective data indicate that under optimal conditions the measured cross-sections are consistent with condensed-phase structures. Seeking a direct measurement, Robinson and co-workers configured a quadrupole time-of-flight (q-ToF) MS with a transmission electron microscopy (TEM) grid holder and deposited ions of GroEL and ferritin[7,8]. These studies imaged particles of roughly the correct size and shape via negative and positive staining TEM. The resultant images, however, lacked the higher resolution features typical of a conventional staining experiment. This lack of detail opens the possibility of structural damage occurring during the experiment and prevents solving the 3D structure from the images. More recently, Longchamp et al. imaged soft-landed small proteins using low-energy electron holography followed by numerical reconstruction—again, confirming the ability to soft land onto a surface. However, given the small size of the molecules imaged and lack of details in the reconstructed images it is difficult to tell whether the landed proteins are damaged or not[9].

Here we describe a modified Orbitrap mass spectrometer system capable of depositing a native MS ion beam onto the surface of TEM grids. With this system and the use of a chemical landing matrix, we demonstrate that non-covalent gaseous ions of protein-protein complexes can retain their condensed-phase structures by obtaining a 3D reconstruction of landed particles. We conclude that this technology and data provide a path towards an integrated mass spectrometry-electron microscopy methodology.

## Results

**Soft-landing enabled Orbitrap mass spectrometer.** Building from the pioneering work described above, we explored new configurations and approaches for depositing gas-phase protein complexes onto surfaces for direct EM imaging. First, we modified a quadrupole Orbitrap hybrid system (Ultra-High Mass Range Q-Exactive[10]) by removing the collision cell and installing an insertion probe to hold a TEM grid to the rear of the c-trap exit lens (Supplementary Figs. 1, 2). With this modification the system can not longer perform HCD; however, the ion beam can either be mass analyzed as usual or directed to a TEM grid.

To test the apparatus, we analyzed the bacterial chaperonin GroEL, an ~800 kDa homo-oligomer having 14 identical subunits, as it has been extremely well-characterized both by native MS and microscopy[11]. Using nano-electrospray ionization we observed charge states ranging from +71 to +62 across the $m/z$ range of 11,000 to 13,000 and having the calculated molecular weight of 802,500 ± 300 Da (Fig. 1a). Next, we deposited the GroEL ion beam for durations of 60–600 s onto glow discharged carbon-coated TEM grids. After deposition, the TEM grids were removed from vacuum, stained with uranyl acetate, and immediately viewed using TEM (Fig. 1b). Particles were observed across much of the TEM grids at reasonably high densities; however, the structural features that define GroEL, i.e., rings and lines, were not present. Instead, irregular shaped, but

mainly featureless particles of approximately the size of GroEL were observed. These images reproduced well those reported by Robinson et al.[7,8]

From these data we hypothesize that the GroEL ions had lost their condensed-phase structure via: (1) the process of ionization, vaporization, and transport through the MS, (2) lengthy exposure to high vacuum without solvent (i.e., up to 600 s on the surface), (3) dissociation upon collision with grid surface during landing, and/or (4) interactions with the TEM grid surface. The most central of these possibilities being the effect of vacuum exposure—about 10 min for the landing durations used in Fig. 1b. To test the effects of vacuum exposure we pipetted GroEL particles directly onto a TEM grid, placed the grid in vacuum (15 min at $2 \times 10^{-5}$ Torr), followed by staining and TEM imaging. The resultant images (Supplementary Fig. 3) contained irregular shaped, featureless particles of approximately the size of GroEL, much like those in Fig. 1b. These data reveal that extended exposure to vacuum alone abolishes the overall structural integrity of the GroEL assembly.

**In vacuo protein preservation.** Cryoprotective compounds (e.g., glycerol, trehalose, glucose, ionic liquids, etc.) can promote preservation of protein structure, even when dehydrated and/or in vacuum environments; further, several studies have shown the benefits of direct TEM imaging from sugar-fixed particles[12–15]. Additionally, two MS studies reported using glycerol-coated deposition surfaces to collect and, ultimately, show biological viability of soft-landed single proteins and intact viruses[16,17]. Following these leads we employed a uniform thin film of a glycerol matrix by depositing a small volume (~3 μL) of glycerol/ methanol (50/50 volume) onto the carbon TEM grid surface followed by edge blotting to remove excess. The coated grid was then spotted with GroEL, placed in a vacuum (10–60 min, $2 \times 10^{-5}$ Torr), and imaged using negative stain TEM. The resultant images revealed structurally intact GroEL (Supplementary Fig. 3) and had no detectable differences than those not exposed to vacuum. Next, grids pretreated with a glycerol matrix were used to perform GroEL landing experiments. Following a deposition period of up to 30 min, the TEM grids were removed and negatively stained. With the glycerol matrix present, similar GroEL particle distributions were observed as in the previous landing attempts; however, the particles displayed the characteristic features expected for negatively stained and structurally intact GroEL (Fig. 1c) as observed in the conventionally prepared sample (Fig. 1d). Note the clearly visible seven-member rings (end views) and the four lines indicating the tetrameric stacked rings (side views) for both the matrix-landed and conventional samples.

To test whether this phenomenon was unique to GroEL, we explored the deposition behavior of two other well-studied protein complexes. The first, alcohol oxidase (AOX, a homo-octamer)[18] was similarly subjected to nano-electrospray and measured at a mass of 598,400 ± 400 Da (Fig. 1e). Deposition of these ions onto the bare TEM grids again resulted in visibly distorted particles of varying size (Fig. 1f). Figure 1g displays an image of these same cations when deposited onto glycerol-treated TEM grids. As with GroEL, the expected structural features of the AOX complex are preserved in the presence of the matrix (conventionally prepared AOX shown in Fig. 1h). Next, we analyzed the tetrameric β-galactosidase complex which comprises four identical 1023 residue long subunits[19]. The smallest of the three complexes studied here, β-galactosidase weighs in at just under 500 KDa (Fig. 1i). Panels **j**, **k** of Fig. 1 illustrate the same trend as observed with GroEL and AOX—the glycerol matrix preserves the landed cationic protein complex.

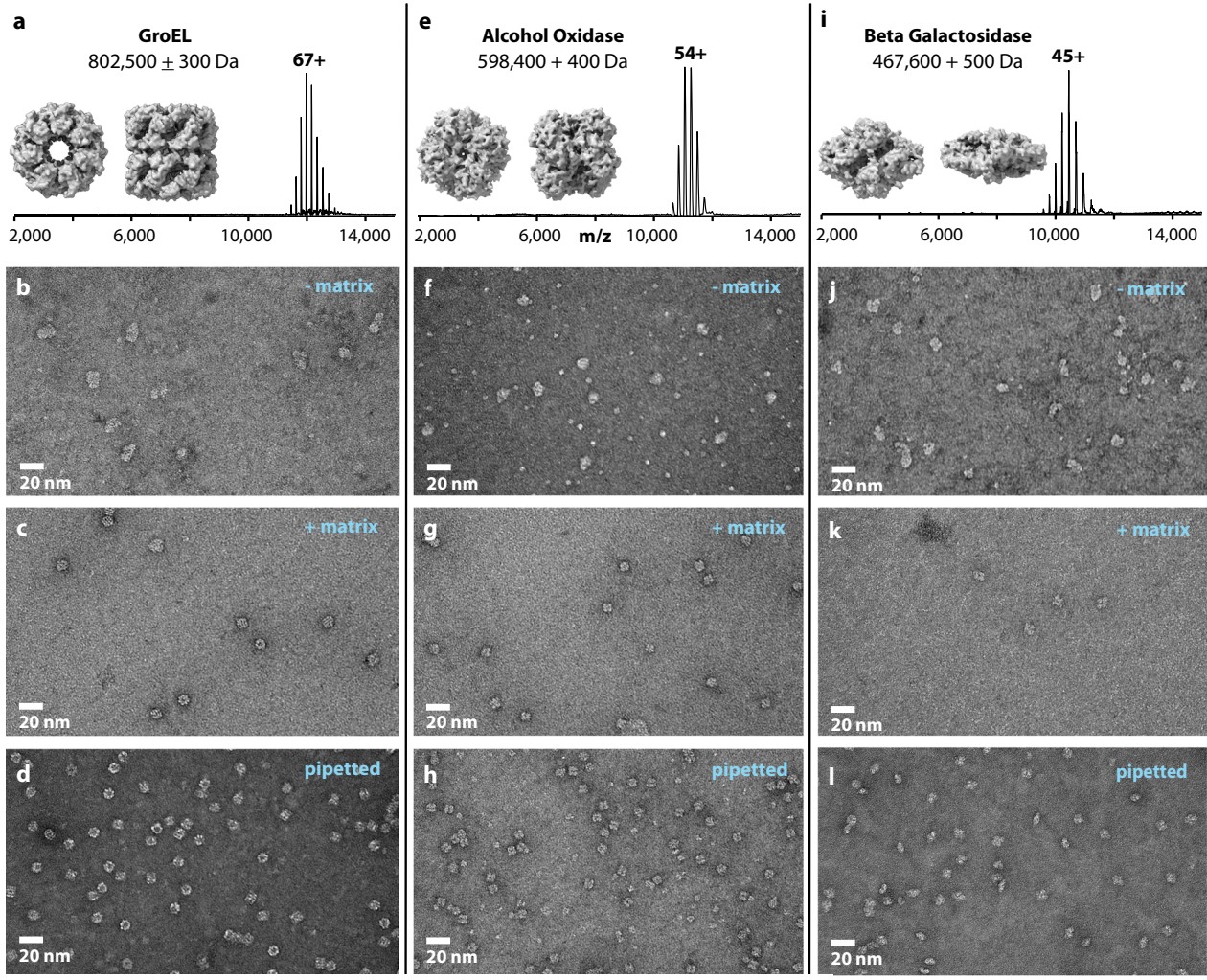

**Fig. 1 Matrix-landed protein complexes retain structural features and integrity as imaged by negative stain TEM. a** Native mass spectrum of GroEL complexes along with molecular model images of GroEL. **b** Negative stain TEM image of GroEL ions landed onto bare carbon TEM grids. **c** Negative stain TEM image of GroEL ions matrix-landed onto TEM grids having a thin film of glycerol. **d** Negative stain TEM image of GroEL molecules that were conventionally prepared. **e** Native mass spectrum of alcohol oxidase complexes along with molecular model images of alcohol oxidase. **f, g** Negative stain TEM images of alcohol oxidase ions landed on bare carbon TEM grids (**f**), or matrix-landing TEM grids coated with a thin film of glycerol (**g**). **h** Conventionally prepared negative stain TEM images of alcohol oxidase molecules. **i** Native mass spectrum the b-galactosidase complex along with molecular model images of b-galactosidase. **j, k** Negative stain TEM images of b-galactosidase ions landed onto either bare carbon TEM grids (**j**) or matrix-landing TEM grids coated with a thin film of glycerol (**k**). **l** Conventionally prepared negative stain TEM images of b-galactosidase molecules. Note the structural models contained in panels (**a, e**, and **l**) were generated from PDB structures (PDB:5W0S, PDB:6H3G, PDB:6X1Q) and included here to aid in image interpretation[21, 26, 27]. Images similar to those shown in panels (**b-l**) were acquired no fewer than five times each.

**Matrix-landing and TEM imaging of GroEL.** Having established a method to deposit and preserve protein complexes from a gaseous ion beam, we sought to probe the decades old question of whether gas-phase biomolecular ions retain their condensed-phase structures. From a GroEL matrix-landed grid we collected a dataset of ~600 images on a Technai G2 Spirit BioTwin microscope equipped with a NanoSprint15 MK-II 15 Mpix camera. We expected that with these matrix-landed molecular images we could generate medium resolution (~20 Å) negative stain 3D reconstruction. We selected ~50 of the highest quality images, having a pixel size of 3.4 Å, and picked and processed ~15,000 particles using cisTEM 2D[20]. The classification was performed and ~7000 (47%) particles contained in the high-quality class averages (Fig. 2a) were carried forward for further refinement. In comparison, 75% of the particles are carried forward when employing conventional methods (see below), indicating that a higher fraction of the matrix-landed particles are damaged.

However, these particles are easily removed during the classification process. Ab-initio reconstruction and auto-refinement, assuming D7 symmetry, resulted in the 3D reconstruction shown in panels **b** and **c** of Fig. 2. Figure 2b displays the superposition of the matrix-landed 3D reconstruction of GroEL (gray) with the high-resolution crystal structure (multi-colored ribbon). These data show excellent agreement between the landed particles and previously determined GroEL structure (PDB:5W0S)[21] in all areas except for two helices which project from the density (Fig. 2b, discussed below). The correlation coefficient between the landed map and a density map simulated from the model with a resolution cut-off of 15 Å was measured as 0.845 using UCSF Chimera[22]. As a further control, we obtained a reconstruction of the same sample prepared conventionally. From this grid we collected a dataset of ~40 images and analyzed them in an identical manner as the matrix-landed sample, with ~9000 particles being picked and ~7000 (75%) being carried forward after

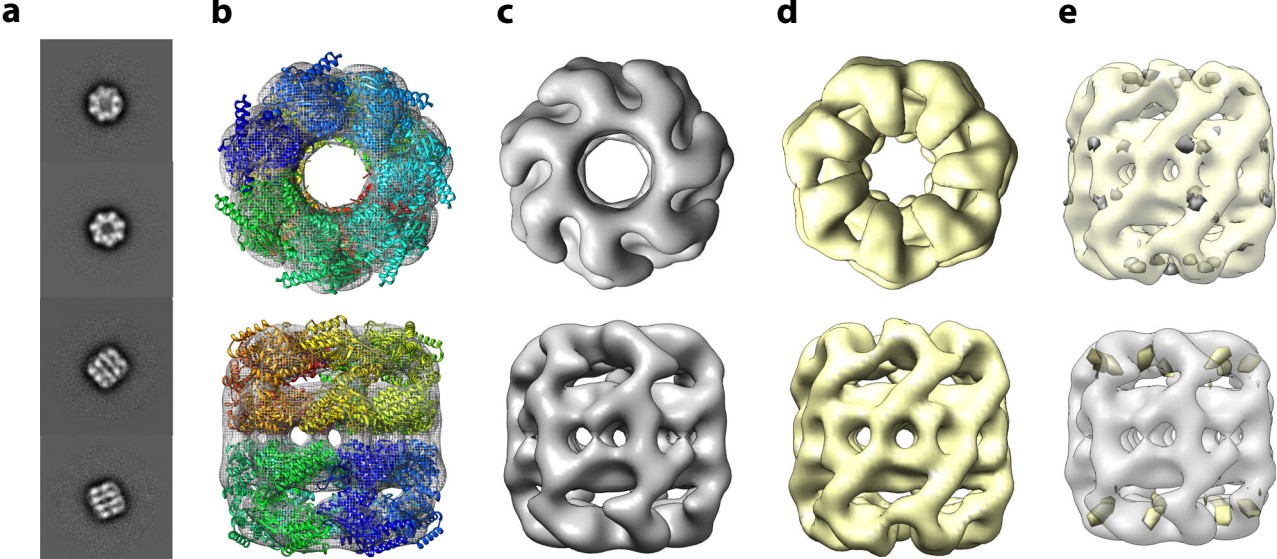

**Fig. 2 3D reconstruction of GroEL complexes that were either matrix-landed from the ion beam of a modified Orbitrap mass spectrometer or conventionally prepared. a** 2D Class averages obtained from negatively stained matrix-landed GroEL cations. **b** Top and bottom views of a three-dimensional reconstruction of GroEL made from the particles contained within the class averages shown in a, fit to a previously determined GroEL structure (PDB:5W0S). **c** Top and bottom views of (**b**) without ribbon model. **d** Top and bottom views of conventionally prepared and imaged GroEL. **e** Difference maps of matrix-landed and conventionally prepared GroEL. Top structure in (**e**) presents the subtraction of the conventionally prepared model from the landed model (where gray indicates difference) while the bottom displays the reverse (yellow indicating difference).

2D classification (Fig. 2d, with a resolution cut-off of 15 Å was measured as 0.912 using UCSF Chimera). Figure 3 closely examines the fit of the atomic model to the single-particle reconstructions of GroEL for a single subunit. Note that density for the aforementioned projecting helices is also absent in the conventionally prepared sample (Fig. 3) suggesting heterogeneity in that part of the structure. Overall, these two reconstructions from either matrix-landed (Fig. 2c) or conventionally prepared GroEL (Fig. 2d) are remarkably similar. Difference maps, shown in Fig. 2e, confirm this, revealing only a few small differences. Whether these differences are bona fide or simply due to local resolution variations between the datasets is presently unclear. Either way, these results provide the highest resolution experimental data collected to date confirming that non-covalent protein complexes subjected to ionization and mass spectrometry can largely, if not completely, retain their condensed-phase structures.

## Discussion

Here we describe a technique, matrix-landing, that promotes the preservation of structure in non-covalent protein complexes that have traversed a mass spectrometer and been deposited onto a TEM grid. With this method we confirm—at the highest resolution to date—that (1) the process of ionization, vaporization, and transport through the MS, which is on the order of ~10 ms in our system, can be accomplished without loss of particle structural integrity and (2) lengthy exposure of particles to high vacuum without solvent/matrix is problematic. Presently we believe that protection from dehydration is the key factor governing the structural preservation of landed particles. That said, we can neither rule the possibility that the matrix may provide additional channels for energy dissipation upon particle impact nor the potential for the matrix to inhibit undesirable surface interactions of the particle with the TEM grid itself. Understanding the precise mechanisms that underlie the preservation promoting abilities of the matrix is a current focus of our efforts. Detailed evaluations of several relevant native MS conditions (*e.g.*, buffer, ion source, desolvation energies, time in transit, etc.)

should also yield further insights and improvements. Exploration of other landing matrices and methods for generating the highest performing matrix films are similarly critical. Landing conditions likewise play an essential role mandating thorough characterization of ion beam energetics, landing pressures, and effects of vacuum exposure.

We suppose that precise control of the TEM grid surface temperature could also be key for the maintenance of the ideal matrix surface and for high-resolution structural preservation. Note all the depositions described herein were conducted at room temperature; moving forward we will explore reduced TEM grid temperatures aiming to both extend the protective effects of glycerol and to aid in the retention of any water/solvent associated with the protein complex ion. The ultimate extension of this reduced-landing temperature concept is to deposit partially hydrated and mass-selected samples directly onto cryogenically cooled (<180 °C) TEM grids. A thin coating of amorphous ice, which could be generated in vacuo, would provide protection from the deleterious effects of both vacuum and the TEM grid surface[23]. Directly coupling cryo-EM grid preparation to MS could provide a host of advantages over conventional grid preparation including improved signal and decreased beam-induced motion (due to lower ice background and lack of in-built ice strain[24], respectively). Aside from the aforementioned benefits derived from MS, the boosted signal could increase resolution and enable imaging of smaller particles. We conclude that the ability to deposit and preserve protein complexes within a vacuum environment may allow for improved TEM sample preparation capabilities and facilitate the integration of two disparate fields into one unified technology.

## Methods

**Materials**. Water (Optima LC/MS grade, W6-4) and methanol (Optima for HPLC, A454SK-4) were purchased from Fisher Chemical. Ammonium Acetate (431311-50G), Glycerol (for molecular biology, G5516-100ml), Amicon Ultra-0.5 centrifugal filter (Ultracel-100 regenerated cellulose membrane, UFC510024). Alcohol Oxidase (Pichia pastoris buffered aqueous solution, 55 mg protein/mL, A2404-1KU), GroEL (Chaperonin 60 from *Escherichia*, C7688-1MG), and β-Galactosidase (from *Escherichia coli*, G3153-5MG) were purchased from Sigma Aldrich. Uranyl

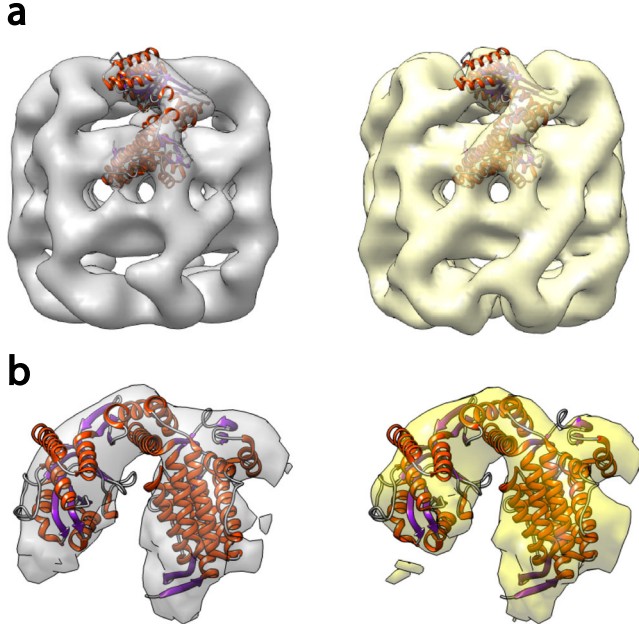

**a**

**b**

**Fig. 3 Single subunit density maps of GroEL from either matrix-landing (left, gray) or conventional samples (right, yellow). a** Fit of a single subunit PDB reference ribbon structure to the overall models obtained by either method. Note the exposed helices in both models that are not contained within the calculated model density. These data confirm that the matrix-landing sample is no different than the result obtained by conventional negative staining TEM. **b** Expanded view of a single GroEL subunit where matrix-landed reconstruction is gray and conventionally prepared is yellow. Ribbon structure taken from PDB:5W0S.

Acetate (1% solution, 22400-1) and TEM grids (Carbon support film on 400 mesh copper, CF400-CU) were purchased from Electron Microscopy Sciences.

**Sample preparation**. GroEL was prepared at 1 mg/mL in 100 mM ammonium acetate. 200 μL of acetone was added to 100 μL of buffered protein solution and allowed to sit for 5 min to precipitate the protein. The sample was centrifuged (Fisherbrand Gusto Mini tabletop centrifuge) and the remaining solvent was removed from the pellet. Following this, 400 μL of buffer was added to redissolve the protein and placed in an Amicon centrifugal filter. The sample was centrifuged (Thermo Scientific Sorvall Legend Micro 21R) at 10,000$g$ for 10 min at 4 °C. A buffer which passed through the filter was discarded and an additional 400 μL of buffer was added to the sample for another round of washing using the same centrifugal settings. To obtain the sample, the filter was inverted and centrifuged at 2000$g$ for 1 min and diluted with 80 μL of buffer.

Alcohol Oxidase was thawed on ice and 10 μL was taken and diluted in 390 μL of buffer. This solution was buffer exchanged by transfer to an Amicon spin filter and centrifuged at 10,000$g$ for 10 min at 4 °C. This cycle was repeated three times replenishing with 400 μL of 100 mM ammonium acetate after each spin. To obtain the final sample the solution remaining above the filter was removed via pipette. β-Galactosidase was prepared at 1 mg/mL in 100 mM ammonium acetate and buffered exchanged as per Alcohol Oxidase. To obtain the final sample, the filter was inverted and centrifuged at 2000$g$ for 1 min and diluted with 200 μL of buffer.

**Mass spectrometry**. All mass spectrometry experiments were performed on a modified Thermo Scientific Q-Exactive UHMR Hybrid Quadrupole-Orbitrap mass spectrometer running Q-Exactive Tune 2.11QF2 control software[10]. Modifications included the removal of the HCD cell ion optics along with the fabrication of a new HCD cell vacuum chamber rear cover plate (see below). Note, without the HCD cell, the system is no longer able to perform collisional activation of mass-selected precursors. Otherwise, the system works as usual and the injection of ions into the Orbitrap is unaffected.

Using readily available components, we implemented a simple device to insert the TEM grid into the vacuum environment of the UHMR mass spectrometer. To insert and remove TEM grids for landing, we adapted the vacuum interlock and probe components of a retired Thermo Fisher Scientific ETD module. Supplementary Fig. 2a presents photographs of these components and their incorporation onto the UHMR. The ion volume insertion and removal tool is used to hold the TEM grid. Incorporation of the inlet valve and guide bar assembly required the fabrication of a new endplate for the vacuum chamber housing of the UHMR HCD cell

(Supplementary Fig. 2b). Supplementary Fig 2c contains the mechanical drawings required to duplicate the endplate. Dimensions are shown in units of millimeters.

The new cover plate (shown in Supplementary Fig. 2b) contained a ball valve assembly which could be evacuated by the roughing pump of the UHMR system. The valve was placed on center with the HCD cell. This arrangement allowed the use of the insertion probe to place and hold a TEM grid at the exit of the c-trap/entrance to the HCD cell without breaking vacuum on the mass spectrometer. No changes were required to the mass spectrometer electronics or software. Borosilicate glass capillaries were pulled in-house using a model P-2000 laser-based micropipette puller (Sutter Instrument, CA) to an emitter inner diameter of 1–5 μm. A platinum wire placed within the capillary provided continuity between the mass spectrometer ESI power supply and the solution being sprayed. All full scan MS1 experiments were conducted with an ESI voltage of 1.1 kV to 1.5 kV, mass resolving power of 6250 at $m/z$ 400, inlet capillary temperature of 250 °C, and in-source trapping with −100V offset. For landing experiments, the Obitrap mass analyzer was not employed, and the ions were not stopped within the c-trap. To prevent trapping, the trapping gas pressure was set to a value of 0.1. A decreasing DC gradient was placed on all the ion optics from the inlet of the mass spectrometer to the TEM grid. Specifically, lens voltages of 20 V, 19 V, 18 V, 17 V, 16 V, 15 V, and 0 V were employed on the injection flatapole, inter flatapole lens, bent flatapole, transfer multipole, C-trap entrance lens, and TEM grid, respectively. All voltages are adjustable through the user interface with exception of the TEM grid which is tied to ground. No insource trapping was employed, the inlet capillary temperature set to 30 °C, and a wide mass filter isolation of 10,000–20,000 $m/z$ for GroEL and 8000–18,000 $m/z$ for Alcohol Oxidase and β-Galactosidase. All protein complex solutions were sprayed at a concentration of approximately 0.1 to 0.3 mg/μL.

**Matrix-Landing**. Plasma-treated carbon film TEM grids were coated with 3 μl of the glycerol/methanol mix (50–50 by volume) and allowed to sit for 30 s. Excess solution was removed by touching the edge of the grid to a piece of filter paper. The remaining solution was allowed to equilibrate for 10 min. The grid was then placed within the mass spectrometer using the insertion probe and exposed to the ion beam for up to 10 min. Upon removal, the grid was negative stained with 75 μL of 1% uranyl acetate by edge blotting. We estimate that the flow rate of our nano-electrospray emitter to be in the range of 20–40 nL/min. For a 10 min deposition experiment we would therefore consume ~300 nL of GroEL solution. At ~0.3 mg/mL; we would consume ~90 ng GroEL.

**Transmission electron microscopy**. TEM studies for the reconstruction of negatively stained GroEL were performed at ambient temperature on a Technai G2 Spirit BioTwin (Thermo Fisher Scientific) fitted with a NanoSprint15 MK-II 15 Mpix camera (AMT Imaging), operated at 120 kV. Microscope and camera control software included Tecnai version 3.1.3 and AMT Capture Engine version 7.00 respectively. For the 3D reconstruction, a defocused image series ranging from 0.1 μm to 2 μm in 0.1 μm steps were collected using the SerialEM software package version 3.8.7 (https://bio3d.colorado.edu/SerialEM/). Collection conditions are summarized in Supplementary Table 1. Particle picking, classification, reconstruction and refinement of 3D maps were performed in cisTEM version 1 (https://cistem.org/). Model fitting, imaging and figure creation were done using UCSF Chimera version 1.16 (https://www.cgl.ucsf.edu/chimera/, Supplementary Fig. 4).

**Reporting summary**. Further information on research design is available in the Nature Research Reporting Summary linked to this article.

## Data availability
GroEL density maps for soft-landed and pipetted samples have been deposited in the Electron Microscopy Data Bank under accession code EMD-26222 [https://www.ebi.ac.uk/pdbe/entry/emdb/EMD-26222][25].

Published structures used for visual comparisons are available in the Protein Data Bank under the accession codes 5W0S [https://doi.org/10.2210/pdb5W0S/pdb] (GroEL), 6H3G [https://doi.org/10.2210/pdb6H3G/pdb] (Alcohol Oxidase) and 6X1Q [https://doi.org/10.2210/pdb6X1Q/pdb] (Beta-galactosidase).

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

## Acknowledgements
We are grateful for support of this project by the National Institutes of Health R35GM118110 grant (to JJC), the Morgridge Institute for Research, and the University of Wisconsin-Madison. We thank Mike Sussman, Lloyd Smith, Yifan Cheng, Jim Wells, Paul Ahlquist, Desiree Benefield, Andy Ottens, and Julia Laskin for helpful discussions. Finally, we thank Elizabeth Wright and Keith Thompson of the UW-Madison Cryo-EM Research Center for access to TEM instrumentation.

## Author contributions
M.S.W. and J.J.C. conceptually designed the research; M.S.W. constructed the instrumentation; M.S.W., K.W.L., A.Z.S., J.M.L., and T.G. performed the experiments; T.G. conducted 3D structural reconstructions; M.S.W., T.G., and J.J.C. wrote the paper.

## Competing interests
J.J.C. is a consultant for Thermo Fisher Scientific. M.S.W., A.Z.S., K.W.L., T.G., and J.J.C. are inventors of intellectual property related to these results. Patent application submitted, Docket No. 338977: 81-21P US, Applicant: Wisconsin Alumni Research Foundation, Inventors: M.S.W., A.Z.S., K.W.L., T.G., and J.J.C., Title: Matrices for Preservation of Biomolecules in Vacuum. The remaining author J.M.L. declares no competing interest.
