## [Peer Review File · Nature Communications]

REVIEWER COMMENTS

Reviewer #1 (Remarks to the Author):

This is an interesting paper which potentially can generate substantial impact in the field. The authors claim that vacuum exposure, as well as the protein-substrate interactions, can lead to a structural modification of proteins deposited via ESI on a carbon coated TEM-grid. They show that the use of a matrix-assisted deposition, namely the use of a glycerol film, can mitigate these effects. Specifically, they suggest that the glycerol film hampers the electric contact of the protein with the conductive substrate. In this way, the protein cannot discharge and can retain a gas-phase structure.

The experiments presented are of high quality and of general interest but the main conclusions are to my current understanding not convincingly covered.

The landing energy of the protein ions is a crucial parameter in such deposition experiments. This aspect is not discussed in the manuscript. The authors should comment on the landing energy of the deposited proteins. The TEM grid is grounded, as such the proteins landing energy is dictated by the kinetic energy of the incoming molecules. What is the average landing energy? Is it small enough to ensure soft landing?

Specifically, the authors should show that the structural alterations of the molecules deposited on the bare carbon coated TEM grid are not due to the impact of the molecule on the substrate via a kinetic energy transfer to the internal degrees of freedom of the molecules. It is indeed possible that the glycerol matrix is actually providing efficient energy transfer channels allowing the molecule to gently land on the surface as described by H-P Cheng et al. (Science 1993, 260, 1304).

Some additional critical points:

The authors claim that a long vacuum exposure affects the quality of the observed particles.

What is the cause of this effect on the proteins structure? Is it due to chains rearrangement promoted by the thermal energy? Or is it related to a slow discharging of the proteins? Can the authors show these data?

According to the authors' interpretation, the glycerol film would prevent the neutralization of the protein upon landing. In this is the case, the sample surface should slowly charge up; do they observe such phenomenon? How thick is the glycerol film be able to prevent the discharge of the molecules for several minutes?

With respect to this point the authors have tested the landing of the proteins on an ionic liquid matrix. Could they specify more about this test? Why were they not able to observe any kind of particles? Should not they have observed particles similar to what reported for the bare carbon coated TEM grid?

Reviewer #2 (Remarks to the Author):

Westphall et al. describe a modified Orbitrap MS system allowing deposition of intact ionized complexes onto carbon coated EM grids. The prepared grids are then negatively stained and imaged using room temperature electron microscopy. The originality of this study resides in the use of coating of the EM grids with a thin layer of glycerol which allows to obtain higher quality particles compared to previous studies for three model protein complexes. Further data collection and image processing shows that the global architecture of GroEL is preserved for the matrix-landed grid and an existing GroEL structure can be fit into the obtained low resolution map.

This study participates in laying ground work for integration of those two techniques, mass spectrometry and electron microscopy. Besides it brings new evidence of the structure preservation of proteins ionized and in gaz phase. However several major points, especially regarding the interpretation of obtained structure should be more rigorously addressed and/or clarified:

1. The obtained maps should be deposited on the EMDB.
2. The preservation of GroEL structure is evaluated by docking of a PDB model into the obtained map. Even if the provided figures seem to show good agreement between the two, more quantitative indicators should be provided such as correlation coefficient between map and model, difference map...
3. The authors compare maps obtained using the matrix-landed grid (matrix-landed map) and using conventional negative stain preparation techniques (reference map). The reference map, obtained using conventional grid preparation techniques, is only showed in supplementary material, without direct comparison with the matrix-landed structure. Again, the comparison should be more rigorous

with superimposition of aligned maps and a difference map to highlight the presence or absence of particle distortion in the matrix-landed grid.

4. Image processing is not described for the reference map. What was the number of particles in the final reconstruction ? What is the resolution estimate for this map ? What was the percentage of particles discarded before reaching the final reconstruction ? This information is indeed important to compare the quality of particles between the two sample preparation methods.

5. One advantage of cryoEM is the small amount of sample required to prepare a grid. What was the protein quantity required to prepare the matrix-landed grid ? Is it comparable to the requirements for conventional sample preparation techniques.

6. It is a shame that some of the data is not shown in the manuscript. It could be added the supplementary material without making the main text more cumbersome.

Reviewer #3 (Remarks to the Author):

This is a technically outstanding study by Grant, Coon and colleagues. It builds on the pioneering work of Siuzdak, Bothner and Robinson to present two key advances in so-called 'preparative' mass spectrometry. First, a novel soft-landing device is implemented on the UHMR platform. Second, matrix landing is demonstrated to be a crucial step to either preserve, protect or restore the native structure of protein ions following their trajectory from electrospray to surface landing through the gas phase.

I highly recommend publication of this work, although the current manuscript still has a few major issues that must be addressed first. Along with some additional minor comments these are outlined below:

1) native MS is referred to as an emerging technique in the abstract, though it has been around for almost three decades. The technique is widely used in academia and biopharma and can hardly be characterized as emerging (even though many in the EM field may still be somewhat unfamiliar with the technique).

2) in the abstract a 'chemical landing matrix' is mentioned. Since no actual 'chemistry' is taking place I do not see the added meaning of 'chemical', it is simply a landing matrix.

3) the authors write that the landing matrix 'both preserves and protects' the native structure of the deposited particles. I would like to suggest a third option, that it 'restores' the native structure by solvating the deposited ions. Either way, the authors do not present any evidence to distinguish between preservation, protection, or restoration and should rephrase this (the 'both ... and ...' construction is not justified).

4) I am struggling with the term 'vaporization' as opposed to 'desolvation', which I believe better captures the physical transition into the gas phase of these ions.

5) The structures shown in Figure 1A are a bit misleading as they are presented with the exact same aesthetic as is common for EM density maps throughout the literature. They may leave the wrong impression of being EM reconstructions rather than surface representations of the PDB models.

6) Page 3 discusses possible reasons for the lack of observed structural features after soft-landing without matrix. Point 3 lists 'interactions with the TEM grid surface' as a possible explanation. I want to propose to split this point into two separate phenomena to distinguish effects from the initial surface collision from those related to surface adsorption itself.

7) Some key observations relating to the effect of the landing matrix on native structure, as discussed on page 4, are listed as 'data not shown'. This is unacceptable; show the data or it didn't happen as far as I'm concerned.

8) The discussion on page 5 relating to the 3D reconstruction of GroEL does not do justice to the pioneering work of Siuzdak and Bother, who already clearly showed that virus particles maintain their integrity and even their replication activity after travelling through the gas phase. It is commendable that the authors reproduce this observation with GroEL, but explicit mention of this earlier work at this point in the manuscript would be fair.

9) The fit of the GroEL structure in the EM reconstruction in Figure 2C is difficult to assess. A zoomed-in fit of just the asymmetric unit would be a lot more informative.

10) the discussion about cryoEM in the final paragraph is frankly a bit far-fetched and wishful. I sincerely hope that preparative MS would bring all these benefits, and I think the discussion should

definitely offer a perspective on applications for cryoEM, but the current discussion is too much hype and advocacy without scientific considerations in my opinion.

11) the EM methods are sometimes described as 'cryoEM' in the methods section and supply information, but only negative stain EM data is presented.

12) It is unclear whether the spectra in Figure 1A are collected with the exact setup that has the grid landing device implemented. Can the authors explicitly confirm that this is the case?

13) Implementation of the grid landing device essentially sacrifices the HCD cell of the setup, but no discussion is given on the resulting limitations/trade off of this choice.

14) the technical description of the grid landing device is too minimal for others to reproduce. As it is presented, the paper simply demonstrates the possibility, but can not serve as a resource to others that wish to implement a similar grid landing device on their own mass spectrometer.

Reviewer #1 (Remarks to the Author):

This is an interesting paper which potentially can generate substantial impact in the field. The authors claim that vacuum exposure, as well as the protein-substrate interactions, can lead to a structural modification of proteins deposited via ESI on a carbon coated TEM-grid. They show that the use of a matrix-assisted deposition, namely the use of a glycerol film, can mitigate these effects. Specifically, they suggest that the glycerol film hampers the electric contact of the protein with the conductive substrate. In this way, the protein cannot discharge and can retain a gas-phase structure.

The experiments presented are of high quality and of general interest but the main conclusions are to my current understanding not convincingly covered.

We thank the reviewer for the positive feedback and support. We have added considerable detail to various aspects of the manuscript in this revision and will specifically respond to the concerns in detail below.

The landing energy of the protein ions is a crucial parameter in such deposition experiments. This aspect is not discussed in the manuscript. The authors should comment on the landing energy of the deposited proteins. The TEM grid is grounded, as such the proteins landing energy is dictated by the kinetic energy of the incoming molecules. What is the average landing energy? Is it small enough to ensure soft landing?

Specifically, the authors should show that the structural alterations of the molecules deposited on the bare carbon coated TEM grid are not due to the impact of the molecule on the substrate via a kinetic energy transfer to the internal degrees of freedom of the molecules. It is indeed possible that the glycerol matrix is actually providing efficient energy transfer channels allowing the molecule to gently land on the surface as described by H-P Cheng et al. (Science 1993, 260, 1304).

The total potential drop from the ion source to the landing grid is 20V. The difference between the final lens element and the landing grid is 15V. Thus, the maximum possible landing energy would be 65 charges x 15V or ~ 975 eV. This, of course, would be the energy of the incoming ions provided there were no damping gases or collisions. There is, however, considerable collisional cooling prior to landing that is accomplished in the c-trap, which is located very near the landing grid. While it is not possible to obtain a precise pressure measurement in the c-trap, it is likely in the mTorr range and can be adjusted using an automated valve. We tuned the system so that when the valve is half open no ions can be detected arriving at the grid (through use of a charge-detector in place of the grid). When the valve is open at 10%, i.e., much less gas, signal can be detected at the grid. This is the setting used for landings. From these experiments we can conclude that the actual landing energies of the GroEL ions is likely much lower than 975 eV. For comparison, Wysocki et al. have demonstrated that it takes ~ 13,000 eV to dissociate GroEL complexes using surface induced dissociation. Even in that experiment a fraction of the GroEL remains intact (note no structural analysis could be done, however).

To further investigate, we performed new experiments to examine the impact of vacuum exposure to GroEL particles. Shown in **Extended Data 3A** are GroEL particles that were pipetted onto a bare TEM grid and immediately stained and imaged. **Extended Data 3B** shows the same particles that were placed under vacuum (comparable to the landing apparatus pressures) for a period of 10 minutes prior to staining. **Extended Data 3B** illustrates that GroEL particles that have not been landed, are damaged following 10 minutes of vacuum exposure. These images produce results strikingly similar to those of landed GroEL particles on bare TEM grids. **Extended Data 3C** displays the same experiment except for (1) the TEM grid was treated

with a thin film of glycerol and (2) the sample was exposed to vacuum for 60 minutes. Even after this extended vacuum exposure, intact GroEL particles were clearly visible. Collectively, these data strongly indicate that the glycerol matrix at the very least is providing GroEL, whether pipetted or landed, protection from vacuum exposure. We have included these new images into the manuscript as **Extended Data 3**.

We appreciate the reviewer's suggestion that perhaps a role of the glycerol matrix is to dampen excess landing energies and therefore promote structural preservation. We believe the data presented above strongly indicates that protection from vacuum is the primary means by which the glycerol matrix provides structural preservation. And, as explained above, with our current instrumentation it is not possible for us to precisely measure landing energies. This is the subject of ongoing investigation and we expect that in follow-up experiments we can comment on this.

Still, because we cannot completely at this time rule out the possibility of energy damping by the matrix, we have updated the sentence outlining the possible mechanisms by which the landing matrix is contributing to particle preservation:

"From these data we supposed that the GroEL ions had lost their condensed-phase structure via: (1) the process of ionization, vaporization, and transport through the MS, (2) lengthy exposure to high vacuum without solvent (i.e., up to 600 seconds on surface), (3) dissociation upon collision with grid surface during landing, and/or (4) interactions with the TEM grid surface."

Some additional critical points:

The authors claim that a long vacuum exposure affects the quality of the observed particles. What is the cause of this effect on the proteins structure? Is it due to chains rearrangement promoted by the thermal energy? Or is it related to a slow discharging of the proteins? Can the authors show these data?

We have addressed this issue of vacuum exposure through the new experiments described above. The mechanisms of structural degradation due to vacuum exposure, beyond dehydration, are not fully known to us and beyond the scope of this communication. It will, however, be the subject of future planned experiments. That said, we believe our new data provides convincing evidence of the role of glycerol to provide protection for both pipetted and landed particles.

According to the authors' interpretation, the glycerol film would prevent the neutralization of the protein upon landing. In this is the case, the sample surface should slowly charge up; do they observe such phenomenon? How thick is the glycerol film be able to prevent the discharge of the molecules for several minutes?

To address this question, we covered the collecting surface of a charge detector that was placed in the location of the grid (mentioned above) with glycerol. Interestingly, the presence of

Extended Data 3. Negative stain TEM images of pipetted GroEL protein complexes onto TEM grids with and without glycerol. (A) GroEL particles pipetted onto bare TEM grids followed by immediate staining and imaging. (B) Same GroEL pipetted particles that were first exposed to vacuum for ten minutes followed by staining and TEM imaging. (C) Same as (B) except particles pipetted onto a glycerol matrix coated TEM grid.

the glycerol film had no impact on the measured signal when a GroEL ion beam was deposited. These experiments revealed that charge was able to make it to the conducting surface of the detector. Thus, we expect that charge would similarly migrate through the glycerol to the grid surface and become neutralized. Based on these experiments, we have removed the discussion of glycerol possibly acting as an insulator. Further elaboration on the ionic liquid experiments is noted below.

With respect to this point the authors have tested the landing of the proteins on an ionic liquid matrix. Could they specify more about this test? Why were they not able to observe any kind of particles? Should not they have observed particles similar to what reported for the bare carbon coated TEM grid?

To address this comment, we further explored the ionic liquid matrix experiments described in the initial draft. Shown in **Rebuttal Figure 1A** is an image of GroEL particles that were placed into a solution of ionic liquid (1-Ethyl-3-methylimidazolium methanesulfonate), and then onto a TEM grid, and finally exposed to vacuum for 60 minutes. Protein fragments and fractured GroEL particles (see pointers) are present; however, no intact GroEL was observed. Although at lower densities, similar fragments were also observed following landings of GroEL onto TEM grids coated with the ionic matrix, (**Rebuttal Figure 1B**). No intact GroEL with preserved structure was present. (pardon our curt annotations in the original manuscript, we wrote “no particles” were observed intending to convey “no structurally intact particles” were observed).

Having now further explored this question and established the inability of this ionic liquid matrix to preserve structure of pipetted GroEL, we have elected to remove the discussion of the ionic matrix. In short, these new data suggest the initial experiments provided no insight into the possible role of the matrix to prevent charge dissipation.

Reviewer #2 (Remarks to the Author):

Westphall et al. describe a modified Orbitrap MS system allowing deposition of intact ionized complexes onto carbon coated EM grids. The prepared grids are then negatively stained and imaged using room temperature electron microscopy. The originality of this study resides in the use of coating of the EM grids with a thin layer of glycerol which allows to obtain higher quality particles compared to previous studies for three model protein complexes. Further data collection and image processing shows that the global architecture of GroEL is preserved for the matrix-landed grid and an existing GroEL structure can be fit into the obtained low-resolution map.

This study participates in laying ground work for integration of those two techniques, mass spectrometry and electron microscopy. Besides it brings new evidence of the structure preservation of proteins ionized and in gaz phase. However several major points, especially

Rebuttal Figure 1. Negative stain TEM images GroEL particles in ionic liquid matrix. (A) GroEL particles in a solution of ionic liquid that was pipetted onto bare TEM grids, exposed to vacuum for 60 minutes, followed by and imaging. (B) Negative stain TEM grid of GroEL particles that had been deposited onto a TEM grid coated with an ionic liquid matrix. Note in both instances the lack of structurally intact GroEL.

and an existing GroEL structure

regarding the interpretation of obtained structure should be more rigorously addressed and/or clarified:

We thank the reviewer for the positive feedback and the confirmation of overall impact of the work.

1. The obtained maps should be deposited on the EMDB.

These maps have been deposited to EMDB. Data availability. See added text below in Methods section:

“GroEL density maps for soft-landed and pipetted samples have been deposited in the World Wide Protein Data Bank (wwPDB, EMD-26222).²⁵”

2. The preservation of GroEL structure is evaluated by docking of a PDB model into the obtained map. Even if the provided figures seem to show good agreement between the two, more quantitative indicators should be provided such as correlation coefficient between map and model, difference map...

3. The authors compare maps obtained using the matrix-landed grid (matrix-landed map) and using conventional negative stain preparation techniques (reference map). The reference map, obtained using conventional grid preparation techniques, is only showed in supplementary material, without direct comparison with the matrix-landed structure. Again, the comparison should be more rigorous with superimposition of aligned maps and a difference map to highlight the presence or absence of particle distortion in the matrix-landed grid.

Figure 2. 3D reconstruction of GroEL complexes that were either matrix-landed from the ion beam of a modified Orbitrap mass spectrometer or conventionally prepared. (A) 2D Class averages obtained from negatively-stained matrix-landed GroEL cations. (B) Top and bottom views of a three-dimensional reconstruction of GroEL made from the particles contained within the class averages shown in A, fit to a previously determined GroEL structure (PDB:5W0S). (C) Top and bottom views of (B) without ribbon model. (D) Top and bottom views of conventionally prepared and imaged GroEL. (E) Difference maps of matrix-landed and conventionally prepared GroEL. Top structure in (E) presents the subtraction of the conventionally prepared model from the landed model (where gray indicates difference) while the bottom displays the reverse (yellow indicating difference).

These two critiques are related and so we respond to them jointly. First, the initial reconstruction that was shown in **Figure 2** was not well controlled. That is, we did not capture images of conventionally prepared and negative stained sample at the time of collection. To allow for the requested direct comparison we have performed a new landing of the GroEL particles and 3D reconstruction alongside a matched control of conventionally prepared GroEL from the same sample. These new data have been implemented into a revised **Figure 2** and related text (presented below). The quality of the newly added reconstruction is a bit lower than what we achieved in the initial submission. Note these data were taken on a different, but available, much older microscope with a camera of lower quality. Still, it is overall the same and addresses all the concerns raised so we have elected to include it here given it is better controlled. We believe these new experiments and revisions wholly address the above critiques.

*“Having established a method to deposit and preserve protein complexes from a gaseous ion beam, we sought to probe the decades old question of whether gas-phase biomolecular ions retain their condensed-phase structures. From a GroEL matrix-landed grid we collected a dataset of ~600 images on a Technai G2 Spirit BioTwin microscope equipped with a NanoSprint15 MK-II 15 Mpix camera. We expected that with these matrix-landed molecular images we could generate medium resolution (~ 20 Å) negative stain 3D reconstruction. We selected ~50 of the highest quality images, having a pixel size of 3.4 Å, and picked and processed ~15,000 particles using cisTEM 2D. Classification was performed and ~7,000 (47%) particles contained in the high-quality class averages (**Figure 2A**)*

*were carried forward for further refinement. Ab-initio reconstruction and auto-refinement, assuming D7 symmetry, resulted in the 3D reconstruction shown in panels B and C of **Figure 2**. **Figure 2B** displays the superposition of the matrix-landed 3D reconstruction of GroEL (gray) with the high-resolution crystal structure (multi-colored ribbon). These data show excellent agreement between the landed particles and previously determined GroEL structure (PDB:5W0S) in all areas except for two helices which project from the density (**Figure 2B**, discussed below). The correlation coefficient between the landed map and a density map simulated from the model with a resolution cut-off of 15 Å was measured as 0.845 using UCSF Chimera. As a further control, we obtained a reconstruction of the same sample prepared conventionally. From this grid we collected a dataset of ~40 images and analyzed them in an identical manner as the matrix landed sample, with ~9,000 particles being picked and ~7,000 (75%) being carried forward after 2D classification (**Figure 2D**, with a resolution cut-off of 15 Å was measured as 0.912 using UCSF*

Figure 3. Single subunit density maps of GroEL from either matrix-landing (left, gray) or conventional samples (right, yellow). (A) Shown here are the fit of a single subunit PDB reference ribbon structure to the overall models obtained by either method. Note the exposed helices in both models that are not contained within the calculated model density. These data confirm that the matrix-landing sample is no different than the result obtained by conventional negative staining TEM. (B) Exploded view of single GroEL subunit where matrix-landed reconstruction is gray and conventionally prepared is yellow. Note ribbon structure from PDB:5W0S.

Chimera). **Figure 3** closely examines the fit of the atomic model to the single particle reconstructions of GroEL for a single subunit. Note that density for the aforementioned projecting helices is also absent in the conventionally prepared sample (**Figure 3**) suggesting heterogeneity in that part of the structure. Overall, these two reconstructions from either matrix-landed (**Figure 2C**) or conventionally prepared GroEL (**Figure 2D**) are remarkably similar. Difference maps, shown in **Figure 2E**, confirm this, revealing only a few small differences. Whether these differences are bona fide or simply due to local resolution variations between the datasets is presently unclear. Either way, these results provide the highest resolution experimental data collected to date confirming that non-covalent protein complexes subjected to ionization and mass spectrometry can largely, if not completely, retain their condensed-phase structures.”

4. Image processing is not described for the reference map. What was the number of particles in the final reconstruction? What is the resolution estimate for this map? What was the percentage of particles discarded before reaching the final reconstruction? This information is indeed important to compare the quality of particles between the two sample preparation methods.

We have added this information, see revised text above.

5. One advantage of cryoEM is the small amount of sample required to prepare a grid. What was the protein quantity required to prepare the matrix-landed grid? Is it comparable to the requirements for conventional sample preparation techniques.

We estimate that the flow rate of our nano electrospray emitter to be in the range of 20 to 40 nL/min (Mann, Analytical Chemistry, 1996, 68 (1), 1-8). For a 10 minute deposition experiment we would therefore consume ~ 300 nL of GroEL solution. At ~ 0.3 mg/mL; we would consume ~ 90 ng GroEL. Note we place about 1 μ L of total GroEL solution into the nanospray needle, but only consume about 90 ng GroEL. For pipetted experiments we deposited approximately 3 μ L of ~ 0.03 mg/mL GroEL solution for about 90 ng total material.

At present, the current amount sampled is comparable to how we are conventionally preparing our negative stain TEM samples. Moving forward we believe it will be possible improve efficiencies of ion transport and deposition so that we reduce overall deposition times. But it is comparable. We have added the estimates above to the methods section so that interested readers could find this information.

6. It is a shame that some of the data is not shown in the manuscript. It could be added the supplementary material without making the main text more cumbersome.

Agreed. We have added several new figures to the manuscript in response to this and other reviewer feedback.

Reviewer #3 (Remarks to the Author):

This is a technically outstanding study by Grant, Coon and colleagues. It builds on the pioneering work of Siuzdak, Bothner and Robinson to present two key advances in so-called 'preparative' mass spectrometry. First, a novel soft-landing device is implemented on the UHMR platform. Second, matrix landing is demonstrated to be a crucial step to either preserve, protect or restore the native structure of protein ions following their trajectory from electrospray to surface landing through the gas phase.

I highly recommend publication of this work, although the current manuscript still has a few major issues that must be addressed first. Along with some additional minor comments these are outlined below:

We thank the reviewer for their support of our work, its potential impact, and agreement in its publication. Below we have addressed each of the specific concerns.

1) native MS is referred to as an emerging technique in the abstract, though it has been around for almost three decades. The technique is widely used in academia and biopharma and can hardly be characterized as emerging (even though many in the EM field may still be somewhat unfamiliar with the technique).

We have removed the word emerging from the manuscript and updated the relevant sentence as follows:

“Native mass spectrometry (MS) is increasingly used to provide complementary data to electron microscopy (EM) for protein structure characterization.”

2) in the abstract a 'chemical landing matrix' is mentioned. Since no actual 'chemistry' is taking place I do not see the added meaning of 'chemical', it is simply a landing matrix.

We elected to use the phrase 'chemical matrix' to convey the concept of adding a chemical – glycerol – onto the grid surface for preservation of deposited particles. We agree that chemical reactions per se are likely not occurring; however, the glycerol is interacting with the landed particles either helping to retain any residual solvent or taking its place, i.e., stabilizing chemical interactions. We used the phrase 'chemical matrix' three times in the original draft and have reduced that to one time in the revised draft. We hope this limited use will be acceptable to the reviewer.

3) the authors write that the landing matrix 'both preserves and protects' the native structure of the deposited particles. I would like to suggest a third option, that it 'restores' the native structure by solvating the deposited ions. Either way, the authors do not present any evidence to distinguish between preservation, protection, or restoration and should rephrase this (the 'both ... and ...' construction is not justified).

This is a valid point – we did not rule out the possibility that the matrix could act to restore structure. New data collected in response to Reviewer 1 does document that the matrix “preserves” GroEL structure during exposure to vacuum. That is, when pipetted GroEL is placed onto a glycerol matrix and then exposed to vacuum the structure remains intact while GroEL that is placed on a bare grid does not. So, the simplest explanation for our matrix landing observations is that something similar is occurring in those experiments. Further, if the glycerol matrix was capable of restoring structure, one would expect that GroEL which was dehydrated (and lost structure) on the carbon grid could be recovered after exposure to glycerol and/or buffer. We have been unable to successfully revive GroEL (structurally) that was exposed to vacuum. **Rebuttal Figure 2** presents the negative stain TEM images of GroEL after vacuum exposure for 10 minutes on an untreated grid. In **Rebuttal Figure 2A** the grid was immediately stained upon removal from vacuum. In **Rebuttal Figure 2B** the grid was covered in a 10% mixture of Glycerol (by volume) and 50mM ammonium acetate and allowed to rest for 15 minutes at ambient conditions before staining. A slight swelling of the particles can be detected in **Rebuttal Figure 2B**, but no particles regained their structure. Given glycerol and/or buffer is

unable to restore the structural integrity of GroEL under ambient conditions in the time frame of these experiments, we believe it is highly unlikely it could do so under vacuum.

Rebuttal Figure 2. Negative stain TEM images GroEL particles after exposure to vacuum. (A) GroEL particles in that were pipetted onto bare TEM grids, exposed to vacuum (10 minutes), followed by negative staining and imaging. (B) Same as (A) except that after removal from vacuum a solution of glycerol/buffer was pipetted onto the grid surface for a period of 15 minutes, followed by negative staining and imaging. These data demonstrate that the glycerol/buffer solution was not able to repair the structural integrity of the GroEL particles.

4) I am struggling with the term 'vaporization' as opposed to 'desolvation', which I believe better captures the physical transition into the gas phase of these ions.

We selected the term vaporization to describe the process of transferring the analyte ions from the condensed phase to the gas-phase. That use is consistent with the definition of the word “vaporize” and the process. This terminology is also used by Fenn in his seminal descriptions of electrospray:

“For the far larger class of species that cannot generally be vaporized without substantial, even catastrophic decomposition, the problem of producing intact ions is much more refractory.”

J. B. Fenn et al. Mass Spectrometry Reviews. 1990

“...not volatile enough to vaporize without decomposition, might become ions in vacuo ready for mass analysis.”

M. Yamashita, J. B. Fenn. The Journal of Physical Chemistry. 1984

5) The structures shown in Figure 1A are a bit misleading as they are presented with the exact same aesthetic as is common for EM density maps throughout the literature. They may leave the wrong impression of being EM reconstructions rather than surface representations of the PDB models.

We acknowledge the concern here about the structures we show for reference in **Figure 1A, E, and I**. The intent is to give the reader an idea of overall particle shape to aid in their analysis of the images contained in the other panels of the figure. We initially used ribbon models of the PDB structures; however, it was difficult to grasp the overall particle shape with these models. Given that, we created the surface representations shown – and they do a much better job of that. It is relatively straightforward to scan down the panels and examine the various images for particles that bear those rough shapes.

To eliminate possible confusion, as the reviewer notes, we have included the following sentence in the **Figure 1** caption.

“Note the structural models contained in panels A, E, and I were generated from PDB structures and included here to aid in image interpretation.”

6) Page 3 discusses possible reasons for the lack of observed structural features after soft-landing without matrix. Point 3 lists 'interactions with the TEM grid surface' as a possible explanation. I want to propose to split this point into two separate phenomena to distinguish effects from the initial surface collision from those related to surface adsorption itself.

Done. We have added the point:

*“(1) the process of ionization, vaporization, and transport through the MS, (2) lengthy exposure to high vacuum without solvent (i.e., up to 600 seconds on surface), (3) **dissociation upon collision with grid surface during landing**, and/or (4) interactions with the TEM grid surface.”*

7) Some key observations relating to the effect of the landing matrix on native structure, as discussed on page 4, are listed as 'data not shown'. This is unacceptable; show the data or it didn't happen as far as I'm concerned.

Based on further experiments as suggested by Reviewer 1, we have removed the discussion of the ionic liquid matrix results and in all other places added any further data to the Extended Data section.

8) The discussion on page 5 relating to the 3D reconstruction of GroEL does not do justice to the pioneering work of Siuzdak and Bother, who already clearly showed that virus particles maintain their integrity and even their replication activity after travelling through the gas phase. It is commendable that the authors reproduce this observation with GroEL, but explicit mention of this earlier work at this point in the manuscript would be fair.

We have more prominently discussed the Siuzdak and Cooks works.

*“Cryoprotective compounds (e.g., glycerol, trehalose, glucose, ionic liquids, etc.) can promote preservation of protein structure, even when dehydrated and/or in vacuum environments; further, several studies have shown the benefits of direct TEM imaging from sugar-fixed particles.¹²⁻¹⁵ **Additionally, two MS studies reported using glycerol-coated deposition surfaces to collect and, ultimately, show biological viability of soft-landed single proteins and intact viruses.**^{16,17} **Following these leads** we employed a uniform thin film of a glycerol matrix by depositing a small volume (~ 3 μ L) of glycerol/methanol (50/50 volume) onto the carbon TEM grid surface followed by edge blotting to remove excess.”*

9) The fit of the GroEL structure in the EM reconstruction in Figure 2C is difficult to assess. A zoomed-in fit of just the asymmetric unit would be a lot more informative.

We have collected new data and updated **Figure 2** per both Reviewer 2 and Reviewer 3 suggestions. This includes performing a new landing and reconstruction that was collected jointly with a matched pipetted samples for direct comparison. The zoom-in fit requested here has been added as a new figure (**Figure 3**). Please see response to Reviewer 2 critique #4 above for more details.

10) the discussion about cryoEM in the final paragraph is frankly a bit far-fetched and wishful. I sincerely hope that preparative MS would bring all these benefits, and I think the discussion should definitely offer a perspective on applications for cryoEM, but the current discussion is too much hype and advocacy without scientific considerations in my opinion.

We have already developed a method for deposition of amorphous ice on TEM grids *in vacuo*. That work is the subject of an issued patent and was presented at the ASMS meeting in November 2021. Thus, the ability to form thin films of amorphous ice *in vacuo* on grids has already been accomplished. We have modified the second sentence of this paragraph to cite these works. What follows that sentence are simply two potential benefits (listed in two brief sentences) of such capability – reduction of ice background and decreased beam-induced motion. Listing the possible benefits of a new technology is customary in the close of an article and these few sentences seem well within the norm for that purpose.

Figure 3. Single subunit density maps of GroEL from either matrix-landing (left, gray) or conventional samples (right, yellow). (A) Shown here are the fit of a single subunit PDB reference ribbon structure to the overall models obtained by either method. Note the exposed helices in both models that are not contained within the calculated model density. These data confirm that the matrix-landing sample is no different than the result obtained by conventional negative staining TEM. (B) Exploded view of single GroEL subunit where matrix-landed reconstruction is gray and conventionally prepared is yellow. Note ribbon structure from PDB:5W0S.

“The ultimate extension of this reduced-landing temperature concept is to deposit partially hydrated and mass selected samples directly onto cryogenically-cooled (< 180 °C) TEM grids. A thin coating of amorphous ice, which could be generated in vacuo, would provide protection from the deleterious effects of both vacuum and the TEM grid surface.”^{22,23}

11) the EM methods are sometimes described as 'cryoEM' in the methods section and suppl information, but only negative stain EM data is presented.

Good catch. We have removed these.

12) It is unclear whether the spectra in Figure 1A are collected with the exact setup that has the grid landing device implemented. Can the authors explicitly confirm that this is the case?

Yes, these spectra were collected with the exact setup that has the landing device implemented after HCD cell removal.

13) Implementation of the grid landing device essentially sacrifices the HCD cell of the setup, but no discussion is given on the resulting limitations/trade off of this choice.

The following statement has been added to the methods section acknowledging the ramifications of HCD cell removal.

“Without the HCD cell, we are no longer able to perform collisional activation of mass selected precursors. Otherwise, the system works as usual and injection of ions into the Orbitrap is unaffected.”

14) the technical description of the grid landing device is too minimal for others to reproduce. As it is presented, the paper simply demonstrates the possibility, but can not serve as a resource to others that wish to implement a similar grid landing device on their own mass spectrometer.

To enable users to easily develop their own landing device the following information, including a new figure (**Extended Data 2**), has been added in the supplemental data section

Extended Data 2. Overview of landing apparatus and Orbitrap UHMR modifications. (A) Photographs of grid insertion probe, vacuum interlock, and modified Orbitrap UHMR. A three-dimensional rendering of the modified vacuum cover is shown in (B) and the machine drawings with dimensional details shown in (C).

*“Using readily available components, we implemented a simple device to insert the TEM grid into the vacuum environment of the UHMR mass spectrometer. To insert and remove TEM grids for landing, we adapted the vacuum interlock and probe components of a retired Thermo Fisher Scientific ETD module. **Extended Data 2A** presents photographs of these components and their incorporation onto the UHMR. The ion volume insertion and removal tool is used to hold the TEM grid. Incorporation of the inlet valve and guide bar assembly required fabrication of a new endplate for the vacuum chamber housing of the UHMR HCD cell (**Extended Data 2B**). **Extended Data 2C** contains the mechanical drawings required to duplicate the endplate. Dimensions are shown in units of millimeters.”*

REVIEWERS' COMMENTS

Reviewer #1 (Remarks to the Author):

The authors have largely addressed my questions and comments.

I am in favor of publication of the paper. I have just one question/comment which could be addressed in the final version of the manuscript.

The authors state that the main effect behind the structural deformation is probably due to dehydration. In the case of ESI deposited proteins, I have thus the following question: does the glycerol prevent the hydrated ions arriving on the surface to lose water, i.e. the gas phase structure of the protein is retained, or might the glycerol allow the protein to refold?

Reviewer #2 (Remarks to the Author):

I thank the authors for answering the reviews.

One last minor point on my side:

The two EM reconstructions are finally calculated with the same number of particles (~7000). However, only 47% of the initial particles were kept in the final reconstruction of the matrix landed sample vs 75% of particles for the conventionally prepared sample, suggesting that a higher fraction of the protein particles are not intact in the matrix landed sample. This apparent decrease of sample quality does not impair structure determination here since "bad" particles can be sorted during image processing but I think it is still worth noting it.

Otherwise my remarks were addressed and I recommend the manuscript for publication.

Reviewer #3 (Remarks to the Author):

All major comments have been addressed. I particularly appreciate the more complete description of the soft landing device in the revised manuscript. I would still recommend that the main text mentions how the HCD cell is sacrificed for the soft-landing device, instead of in the materials/methods as the authors have done now.

The final perspective/discussion on applications in cryoEM now refer to ASMS proceedings. This is appreciated, though that information is neither publicly accessible, nor peer reviewed. I leave it up to the editor to decide what the journal finds suitable and appropriate in this respect.

I recommend publication and I would like to congratulate the authors on an outstanding piece of work.

Second Round of Review Reponse

Reviewer #1 (Remarks to the Author):

The authors have largely addressed my questions and comments.

I am in favor of publication of the paper. I have just one question/comment which could be addressed in the final version of the manuscript.

The authors state that the main effect behind the structural deformation is probably due to dehydration. In the case of ESI deposited proteins, I have thus the following question: does the glycerol prevent the hydrated ions arriving on the surface to lose water, i.e. the gas phase structure of the protein is retained, or might the glycerol allow the protein to refold?

We thank the reviewer for supporting publication of our manuscript. This is an interesting question that perhaps the glycerol is helping the protein complex refold. While we do not believe this is likely, it is not possible for us to know the exact mechanisms of how the protection is occurring at this time. We believe these complexes are landing, at least partially hydrated, and that the glycerol is aiding in their preservation, perhaps by helping to retain this hydration layer or behaving in a similar manner as the hydration layer. This interpretation is also supported by the extensive ion mobility measurements that indicate protein complexes have the approximate correct shape while transiting the vacuum chamber as gaseous ions.

Reviewer #2 (Remarks to the Author):

I thank the authors for answering the reviews.

One last minor point on my side:

The two EM reconstructions are finally calculated with the same number of particles (~7000). However, only 47% of the initial particles were kept in the final reconstruction of the matrix landed sample vs 75% of particles for the conventionally prepared sample, suggesting that a higher fraction of the protein particles are not intact in the matrix landed sample. This apparent decrease of sample quality does not impair structure determination here since "bad" particles can be sorted during image processing but I think it is still worth noting it.

Otherwise my remarks were addressed and I recommend the manuscript for publication.

We thank the reviewer for their support of the manuscript. We have added the following line to the main text indicating the presence of damaged particles in the landed sample as suggested:

"Note these data indicate that a fraction of the landed particles are damaged; however, they are easily removed during the classification process."

Reviewer #3 (Remarks to the Author):

All major comments have been addressed. I particularly appreciate the more complete description of the soft landing device in the revised manuscript. I would still recommend that the main text mentions how the HCD cell is sacrificed for the soft-landing device, instead of in the materials/methods as the authors have done now.

The final perspective/discussion on applications in cryoEM now refer to ASMS proceedings. This is appreciated, though that information is neither publicly accessible, nor peer reviewed. I leave it up to the editor to decide what the journal finds suitable and appropriate in this respect.

I recommend publication and I would like to congratulate the authors on an outstanding piece of work.

We thank the reviewer for their support of the manuscript. We have added the following line to the main text indicating that you can no longer do HCD without the HCD cell:

“With this modification the system can not longer perform HCD; however, the ion beam can either be mass analyzed as usual or directed to a TEM grid.”

We also have removed the citation to the ASMS proceedings.